# Hydrodynamical effect of parallelly swimming fish using computational fluid dynamics method

**Keisuke Doi[1], Tsutomu Takagi[2], Yasushi Mitsunaga[1], Shinsuke Torisawa[1]***

**1** Graduate school of Agriculture, Kindai University, Nakamachi, Nara, Japan, **2** Faculty of Fisheries Sciences Graduate school of Fisheries Sciences Hokkaido University, Minato, Hakodate, Hokkaido, Japan

* ns_torisawa@nara.kindai.ac.jp

**Data Availability Statement:** All relevant data are within the manuscript.

**Funding:** a Grant-in-Aid for Scientific Research (C) from JSPS, Japan [grant number 20K04355]

## Abstract

Fish form schools because of many possible reasons. However, the hydrodynamic mechanism whereby the energy efficiency of fish schools is improved still remains unclear. There are limited examples of fish models based on actual swimming movements using simulation, and the movements in existing models are simple. Therefore, in this study, we analyzed the swimming behavior of Biwa salmon (*Oncorhynchus* sp., a salmonid fish) using image analyses and formulated its swimming motion. Moreover, computational fluid dynamics analysis was carried out using the formulated swimming motion to determine the fluid force acting on the fish body model with real fish swimming motion. The swimming efficiency of the fish model under parallel swimming was obtained from the calculated surrounding fluid force and compared for different neighboring distances. The flow field around the fish model was also examined. The swimming efficiency of two fish models swimming parallelly was improved by approximately 10% when they were separated by a distance of 0.4L, where L is the total length of the model. In addition, the flow field behind the fish body was examined under both inphase and antiphase conditions and at inter-individual distances of 0.8L and 1.2L. The apparent flow speed in the distance range of 0.5–2.0L from the midpoint of the snouts of the two individuals was lower than the swimming speed. The pressure distribution on the fish model showed an elevated pressure at the caudal fin. Interestingly, we obtained an isopleth map similar to that of a caudal peduncle. To avoid a negative thrust, the aft part of the body must be thin, as shown in the isopleth map obtained in this study.

## Introduction

Many fish form schools, for reasons such as avoidance of predators [1], higher feeding efficiency, and better reproductive opportunities [2]. Various studies have been conducted on fish shoaling behavior and the relationships between individuals. From a hydrodynamical viewpoint, swimming in schools improves the energy efficiency compared to swimming individually [3,4]. To understand the hydrodynamic mechanism underlying the high energy efficiency of this type of swimming behavior, Weihs [5] conducted a pioneering work on the energy efficiency of schooled fish in terms of the vorticities generated in the wake region via the potential theory. Regarding

**Competing interests:** The authors have declared that no competing interests exist.

fish migration mechanisms, Lighthill [6] quantitatively assessed the swimming efficiency of aquatic organisms. Wu [7] discussed fish propulsion mechanisms from a hydrodynamic perspective, whereas Webb [8] attempted to explain fish migration mechanisms from both hydrodynamic and physiological perspectives. Using a pair of swimming fish robots, Li et al. [9] hypothesized that energy savings could be achieved using vortex phase-matching.

With the development of computer technology, computational fluid dynamics (CFD) has increasingly been applied to study the swimming mechanisms of aquatic organisms. In this method, a space is divided into many small elements, and the basic fluid equations for each element are solved numerically to obtain an approximate solution. Liu et al. [10] described the propulsive mechanism of tadpole migration using CFD analysis. Adkins and Yan [11] performed CFD analysis of a fish model in a viscous fluid. Takagi et al. [12] quantitatively evaluated the energy-saving swimming behavior of bluefin tuna (*Thunnus orientalis*) using CFD analysis. Many studies have investigated the hydrodynamic interaction mechanisms for energy saving in fish swimming side by side [13–16]. Hemelrijk et al. [17] applied a particle simulation (multi-particle collision dynamics) method to elucidate the increased energy efficiency of a 2D model when swimming in a school.

Regarding schools of live fish, the European sea bass (*Dicentrarchus labrax*) expends less energy during schooling than during solo swimming [18]. However, it is not possible to isolate and measure the force acting on a fish body in detail in a live fish experiment. Dabiri et al. [19] used an algorithm to determine the pressure distribution from the velocity field to obtain the pressure coefficient. Hemelrijk et al. [17] used multi-particle collision dynamics (MPCD) to estimate the thrust and drag forces acting on a fish body model. Limited work has been done to determine the thrust and drag forces acting on a fish body model using these methods.

In this study, we derived the swimming motions of salmonids from sample fish and applied the motions to a 3D rectangular plate fish body model in order to evaluate the pressure distribution on the lateral surface of the body. The National Advisory Committee for Aeronautics (NACA) 4-digit series was used as the wing section of the fish body model. This is a vertically symmetrical wing type, which has been used as a fish body model in previous studies [20]. Eels are anguilliform, meaning that the entire body moves in waves; tuna are tunniform, meaning that only the tail vibrates; salmonids are subcarangiform, meaning that half of the body moves in waves [21].

The purpose of this study was to clarify the effect of parallel swimming in a group of fish by evaluating the flow field and swimming efficiency around the fish body using CFD analysis from a hydrodynamics viewpoint. Hiraishi et al. [22] analyzed the swimming motion of rainbow trout (*Oncorhynchus mykiss*, a salmonid) and expressed it as a mathematical formulation. Videler and Hess [23] examined the swimming movements of saithe (*Pollachius virens*) and mackerel (*Scomber scombrus*). We analyzed the images of swimming motions of salmonids. By formulating the swimming motion, we calculated the fluid force acting on the fish body model by CFD analysis. There are several examples of body wave functions applied to CFD, but their lateral excursion is simple. The body wave functions used had amplitudes such that the simple envelopes became larger as they approached the tail tip [16,17]. To confirm the effect of parallel swimming on the swimming efficiency, we calculated the swimming efficiency by extracting the drag and thrust forces acting on the fish model, and compared them for different neighboring distances; the flow field around the fish was also examined.

## Materials and methods

This study was specifically approved by Kindai University Committee on Animal Research and Bioethics (Permit Number: KAAG-30-001) and was carried out in strict accordance with the Guiding Principles for the Care and Use of Research Animals.

**Table 1. Test fish parameters.**

| TL (cm) | FL (cm) | BL (cm) | BW (kgf) | Swimming Speed (BL s⁻¹) | | | | | Date of Capture |
|---------|---------|---------|----------|------|------|------|------|------|-----------------|
| 44.9 | 42.8 | 39.7 | 0.806 | 1.01 | 1.26 | 1.52 | 1.76 | 2.02 | August 31, 2015 |

TL, total length; FL, fork length; BL, body length; BW, body weight.

## Formulation of swimming motion

We caught a Biwa salmon (*Oncorhynchus* sp.) in Lake Biwa, Shiga Prefecture, Japan, and used it as the one test fish (Table 1) in our study. The test fish was allowed to swim at one point in a circulating tank (PT-110 (modified), volume: 0.396 m³, observation section: 1.1 m wide, 0.3 m deep, 0.2 m high, West Japan Fluid Engineering Laboratory Co., Ltd.), and it was recorded from the top of the tank using a digital video camera (HDC-TM750, Panasonic Corporation) at 60 fps.

We obtained time-series 2D positional coordinates of 13 points on the fish body from the snout to the tip of the caudal fin based on the filmed video data. The coordinates were extracted using software developed in the laboratory. The points on the body axis were set from the points on the fish body where the position could be identified from the images. Each point was assigned to the snout (a), eyes (b, c), base of the pectoral fin (d, e), posterior end of the pectoral fin (f, g), root of the dorsal fin (h), posterior end of the dorsal fin (i), root of the adipose fin (j), root of the caudal fin (k), or posterior end of the caudal fin (l, m). Moreover, we considered the midpoints of b–c, d–e, f–g, and l–m as points on the body axis, namely n, o, p, and q, respectively (Fig 1). For the posterior end of the caudal fin, the midpoint between the upper and lower ends of the caudal fin was set as the position of the caudal fin tip.

The amplitudes were calculated along the body axis from the 2D positional coordinates of each point on the body axis, and the frequencies and wavelengths of the traveling waves on the body axis were determined.

Hiraishi et al. [22] expressed the swimming motion of rainbow trout as follows (Eqs 1–3).

$$h(x, t) = f(x) \cdot g(x, t) \tag{1}$$

$$f(x) = ae^{bx} + ce^{dx} \tag{2}$$

$$g(x, t) = \cos\left\{2\pi\left(ft - \frac{x}{\lambda}\right)\right\} \tag{3}$$

where $h(x, t)$ is the displacement in the direction perpendicular to the direction of travel of the cross-section of the fish body; where $x$ (BL) is the distance from the snout over the body length that is parallel and opposite to the direction of travel of the fish; $t$ (s) is the time, $f$ is the frequency, and $\lambda$ (BL) is the wavelength of the traveling wave. In addition, $a$, $b$, $c$, and $d$ are the parameters representing the shape of the swimming motion. The relationship between the position $x$ and amplitude on the body axis can be expressed as the sum of two exponential functions: $ae^{bx}$ and $ce^{dx}$. The thrust force of swimming is mainly generated by the vibration of the tail, the part represented by $ce^{dx}$, and because the tail is slightly affected by the amplitude near the head while the amplitude near the head is affected by the amplitude of the tail, $ae^{bx}$ was calculated from the value of $x$ in $ce^{dx}$ minus the value obtained by substituting the value near the head [22]. To obtain the unknown parameters $c$ and $d$ in Eq 2, the amplitude values for $x > 0.4$ were logarithmically transformed and linearly approximated using the least-squares method. In the part where $x < 0.25$, the values of $ce^{dx}$ at the point where $x < 0.25$ were

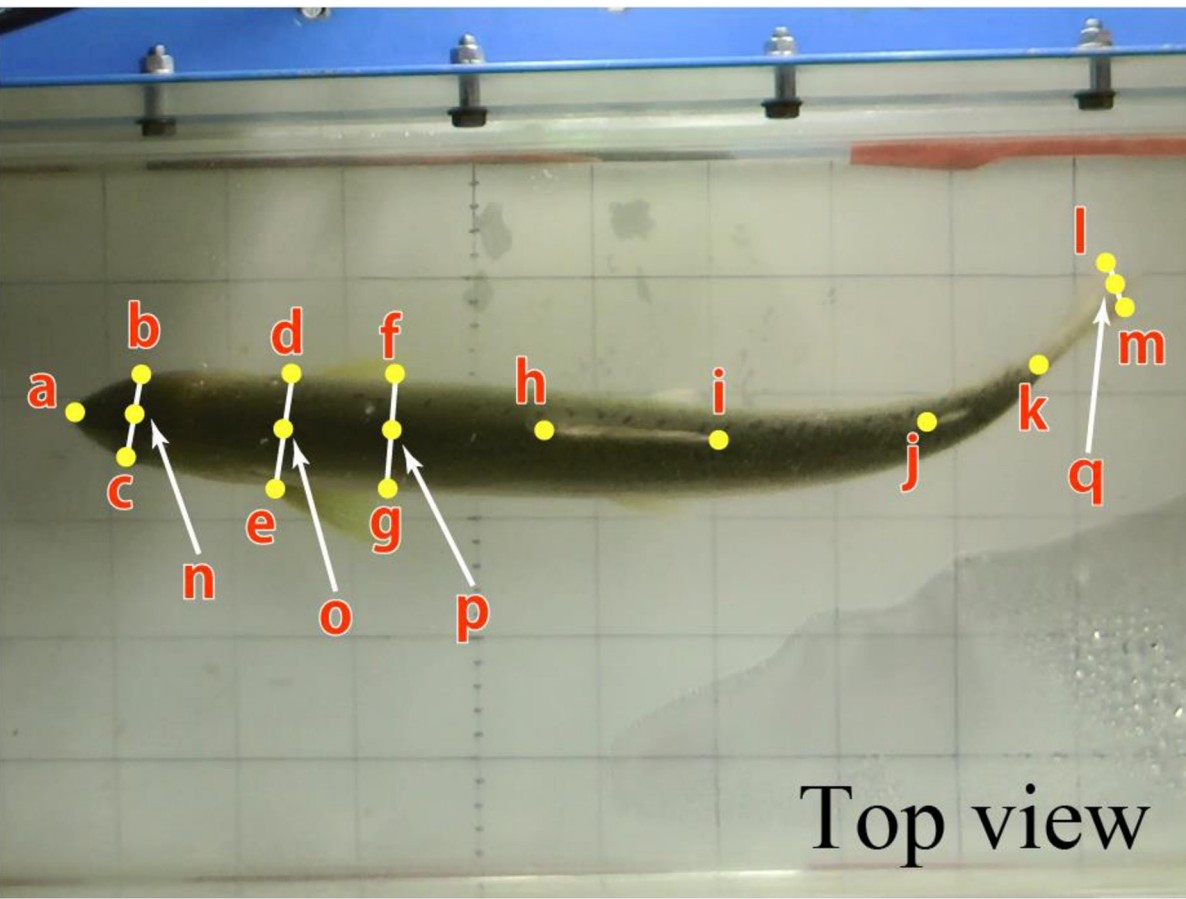

**Fig 1. Points on the body axis.**

obtained, and the unknown constants *a* and *b* in Eq 2 were estimated by subtracting these values from the amplitude values and approximating them in the same manner.

## CFD analysis of solitary swimming

The model used in the CFD analysis was a rectangular plate model with symmetrical wings. The length (L), height, and maximum wing thickness of the fish model were set to 43.9, 8.93, and 4.51 cm, respectively, based on the NACA 4-digit series wing (Fig 2), so that they are equal to the measured data of the Biwa salmon. 3D computer-aided design (CAD) software (Auto-CAD 2017, Autodesk) was used for model creation.

The Navier–Stokes equation of motion for an incompressible viscous fluid can be expressed as follows (Eqs 4 and 5):

$$\frac{\partial U}{\partial t} + (U \cdot \nabla)U = F - \frac{1}{\rho}\nabla p + \frac{\mu}{\rho}\nabla^2 U \tag{4}$$

$$\nabla \cdot U = 0 \tag{5}$$

The solution to these equations can be obtained by satisfying the continuous equations (Eqs 4 and 5). In the equation, *U* is the velocity vector, *t* is time, *F* is the force vector per unit volume, $\rho$ is the fluid density, *p* is the pressure, and $\mu$ is the viscosity coefficient.

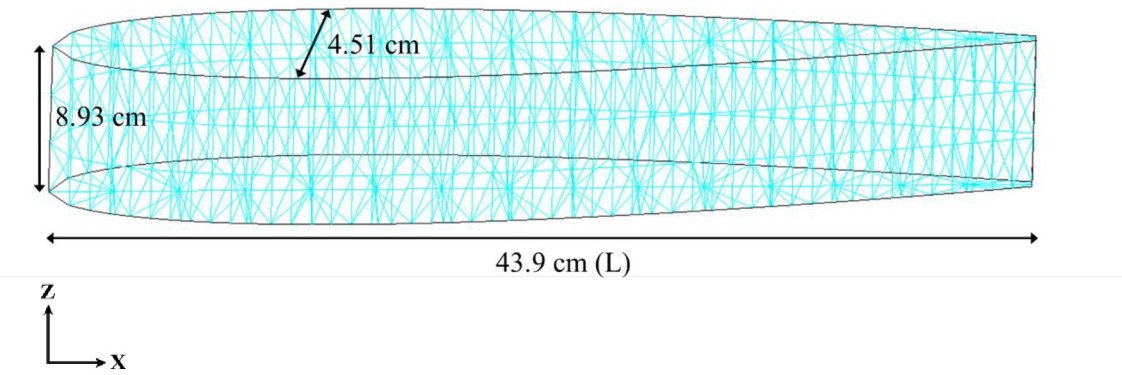

**Fig 2. Model used in the CFD analysis.**

In this study, a 3D thermo-fluid analysis system (SCRYU/Tetra V14, Software CRADLE CO., LTD.) was used to numerically solve the Navier–Stokes equation and continuity equation via the finite volume method and to calculate the physical quantities.

We used the swimming motion function derived from the actual swimming motion of a Biwa salmon as a condition for the movement of elements in the fish model. The element movement condition is a condition for moving the contacts that make up the elements in a movement vector with respect to the volume domain using the arbitrary Lagrangian–Eulerian (ALE) method [24] when performing fluid analysis of moving objects or objects with varying boundary shapes. In the present analysis, the CFD analysis was performed by moving the nodal points of each element in the fish body model during the swimming motion using the derived swimming motion function.

The following conditions were set for the CFD analysis of solo swimming. The analysis area was set to 5.0L in the axial direction, 4.0L in the lateral direction, and 2.0L in the vertical direction, considering the shape change of the fish model from left to right, the view of the backwaters, and the influence of the wall. L is the total length of the fish model.

A steady-state current with five levels of flow was inflowed from the front face of the fish model and outflowed from the rear face with constant pressure and flow direction. The inflow velocities were the same as the velocities used in the live fish experiments, and the swimming motion functions were calculated based on the swimming motion at each velocity recorded in each of the live fish experiments (Fig 3).

It was assumed that the wall conditions were free slip and unaffected by frictional stress for all the surfaces except for the inflow and outflow planes in the analysis region. The fluid temperature, fluid density, viscosity coefficient, and thermal conductivity were assumed to be 20˚C, 998 kg m$^{-3}$, $1.02\times10^{-3}$ (Pa·s), and zero, respectively. The Reynolds number and Strouhal number of the 3D fish model were $2.60\times10^{5}$ and 0.34, respectively. To account for the effects of turbulence, the AKN k–ε model [25] was used, which is suitable for the analysis of unsteady turbulence in the low-to-high Reynolds number range.

In a CFD analysis, the analysis area is divided into small elements, and the solution is obtained by solving the fluid equations for each. Fig 4 shows the mesh partitioning in our CFD analysis. The tetra element size was in the range of 2–64 mm, and the size of the model nearest neighbor element was divided such that the tetra element size was 2 mm so that it became finer when approaching the fish body model. The total number of elements in the analysis area was $1.20\times10^{6}$. In the actual flow, a flow velocity gradient is created between the fluid and the wall due to the frictional stress from the wall, and a thin layer of flow is formed, called the boundary layer. By inserting a layer parallel to the surface of the fish, called the boundary layer

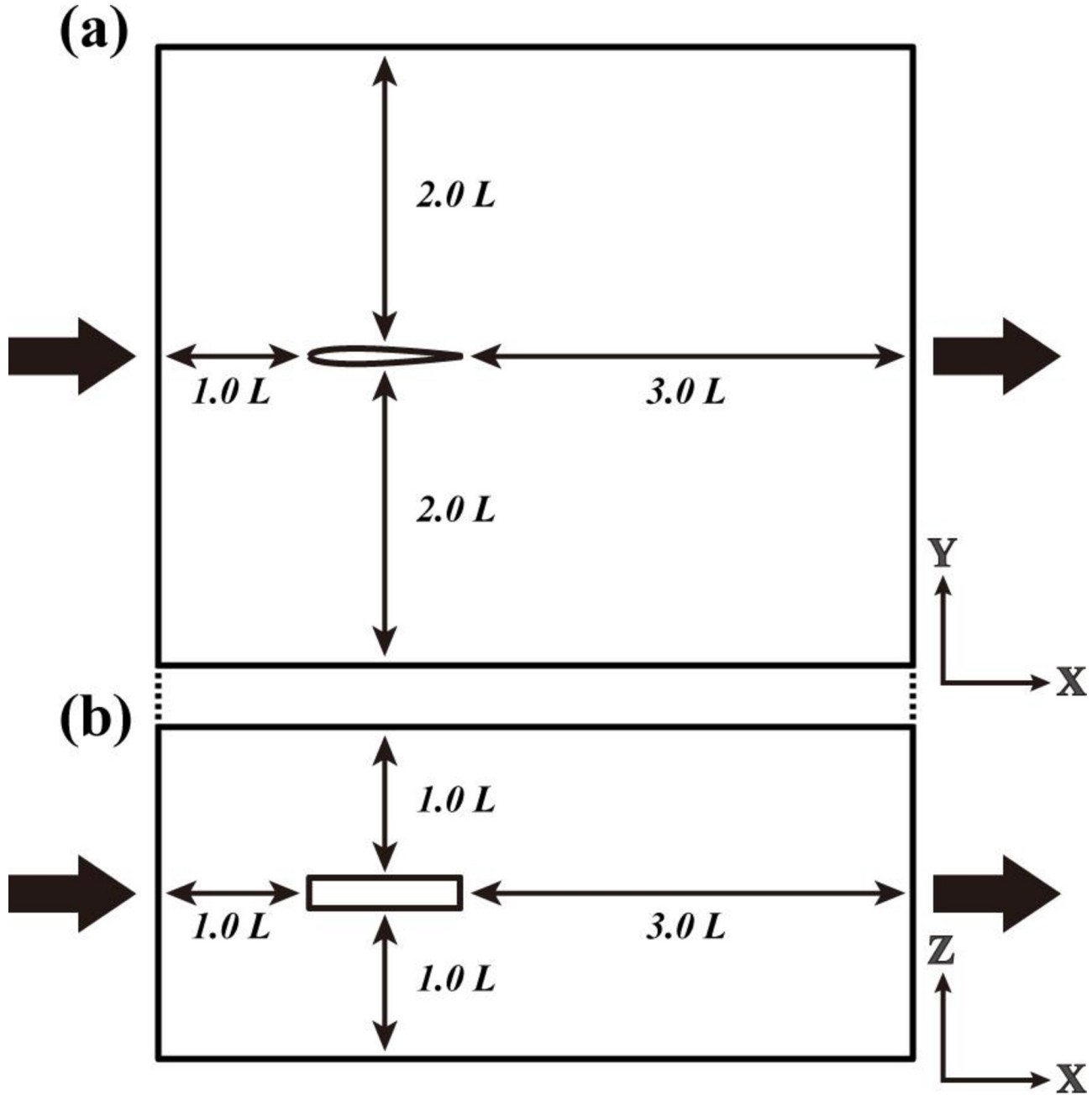

**Fig 3.** Analytical area for CFD analysis of solitary swimming: (a) Top view and (b) side view.

element (prism layer), which represents the gradient of the flow velocity in this boundary layer, we can increase the accuracy of the turbulence analysis. The boundary layer elements were inserted to cover the surface of the fish model in three layers at a rate of change of 1 with a thickness of 0.4 mm for the first prism layer.

In the CFD analysis, the fluid forces acting on each element on the model surface can be converted into arbitrary components. Therefore, it is possible to calculate the fluid force acting on the surface of the fish body model, with the component in the forward direction as the

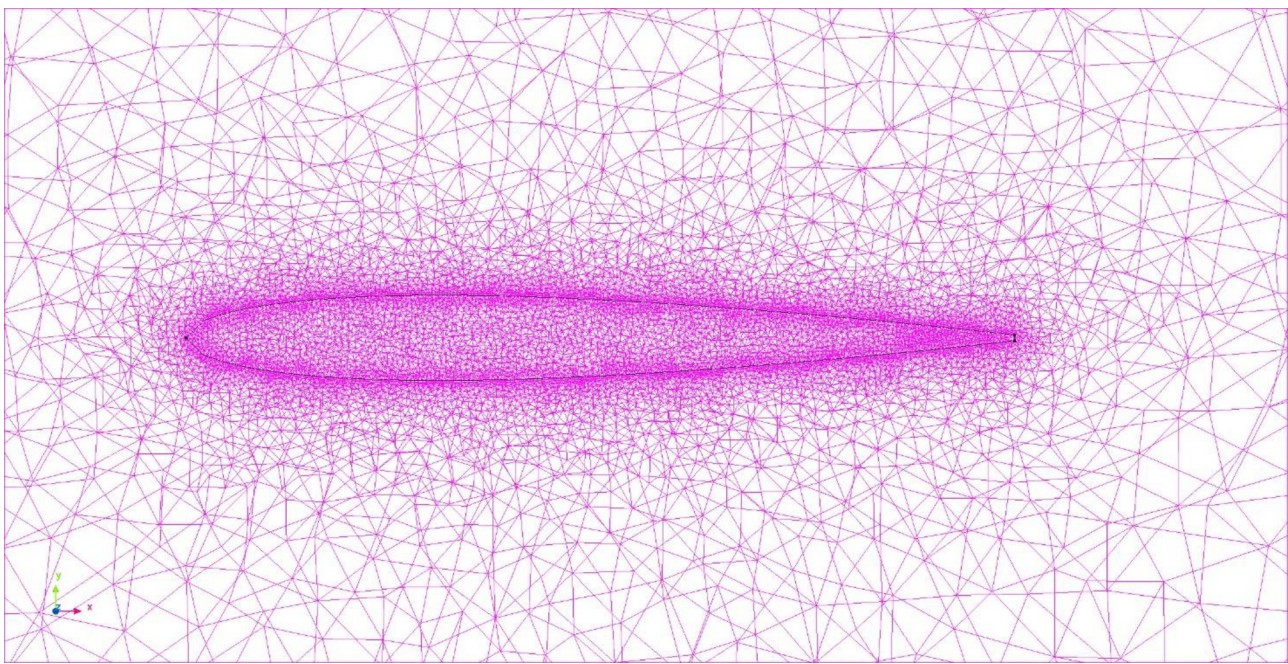

**Fig 4. Mesh partitioning in CFD analysis of solitary swimming.**

thrust force and the opposite component as the drag force acting on the swimming fish body model. In this analysis, among the fluid forces acting on the fish model in the body axial direction, the thrust force was defined as the thrust direction component of the pressure. The drag force was defined as the sum of the drag direction component of the pressure and the frictional drag force.

## CFD analysis of parallel swimming

Two parallel fish models were set up in the analysis area to observe the changes in the backwaters when they migrated in parallel. The polymerized grid method was used to avoid calculation divergence due to the deformation of the mesh during the motion of the two individuals. In an overlapping lattice method, a single analysis space is represented by superimposing a mesh called a dependent region on a mesh called an independent region. The advantage of this method is that it is possible to set the movement conditions for each region and analyze complex objects in terms of their movement and collision.

The model used for the parallel swimming was a rectangular plate model with the same cross-sectional shape of symmetrical wings as the model used for the solo swimming. For the swimming motion, the swimming motion function applied was the same as that applied to the solo swimming.

The conditions for the CFD analysis were as follows. The analysis area was large enough to consider the left and right motions of the fish body model (Fig 5). The inflow velocities used in the analysis were the ones determined from the CFD analysis of solo swimming. At this time, the frequency of caudal fin oscillation was varied so that the thrust and drag forces were largely balanced under each analysis condition. It was assumed that the wall conditions were free slip and unaffected by frictional stress for all the surfaces except for the inflow and outflow planes in the analysis region. The fluid temperature, fluid density, viscosity coefficient, and thermal conductivity were assumed to be 20°C, 998 kg/m$^3$, $1.02 \times 10^{-3}$ (Pa·s), and zero, respectively.

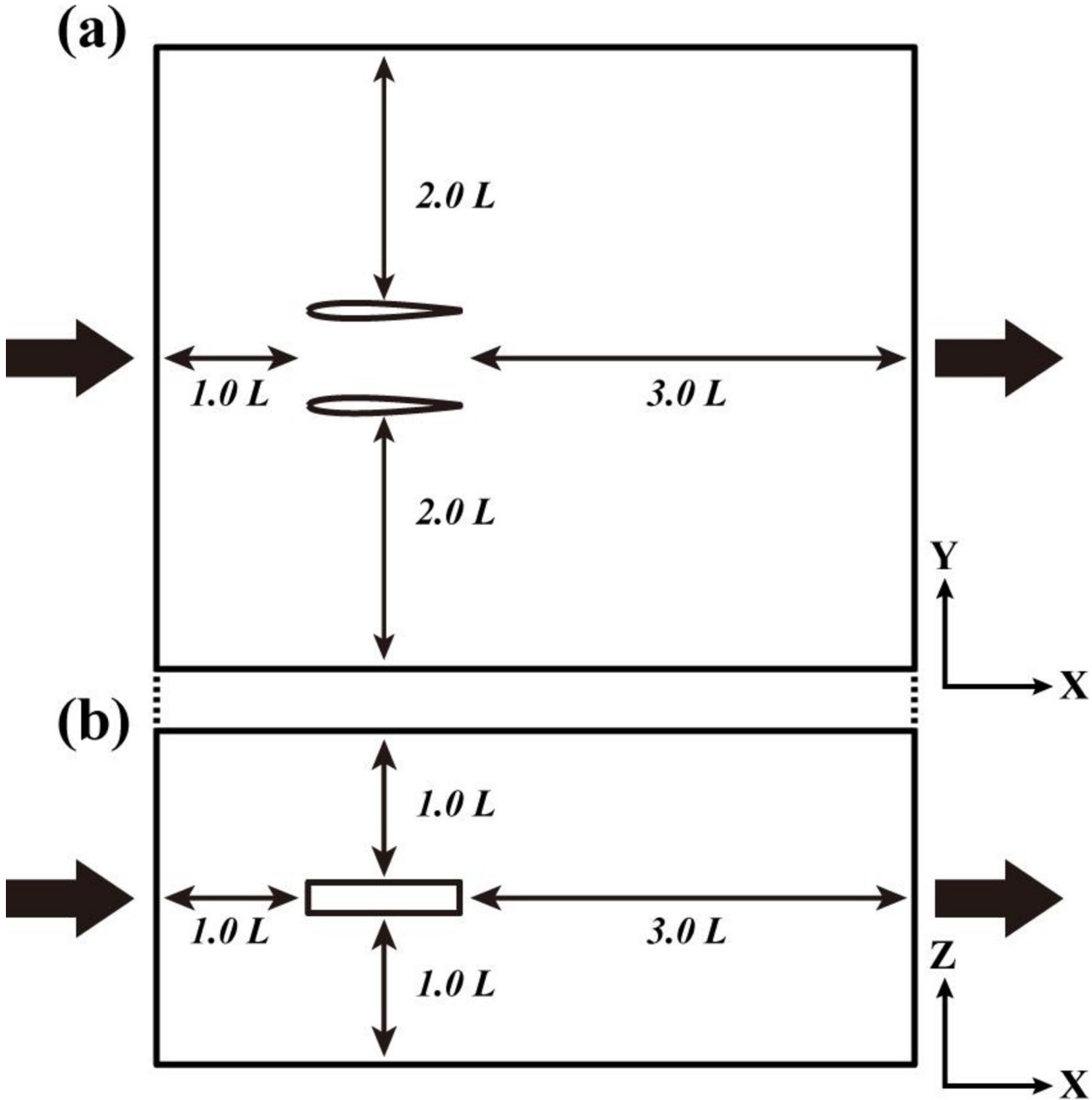

**Fig 5.** Analytical area for CFD analysis of parallel swimming: (a) Plan view and (b) side view.

The Reynolds number of the 3D fish model was $2.60 \times 10^5$. The Strouhal number of the 3D fish model was 0.42 to 0.44. The AKN k–ε model [25], which is suitable for analyzing unsteady turbulence in the low-to-high Reynolds number range, was used as the turbulence analysis model.

To compare the distances between individuals in the parallel swimming model under the same analytical conditions from close proximity to solo swimming, we set the distances to 0.4, 0.6, 0.8, 1.2, 1.6, and 2.0L, and analyzed the cases where the amplitude motion phases were inphase and antiphase, respectively. The inlet flow speed was set to 1.52 BL s$^{-1}$.

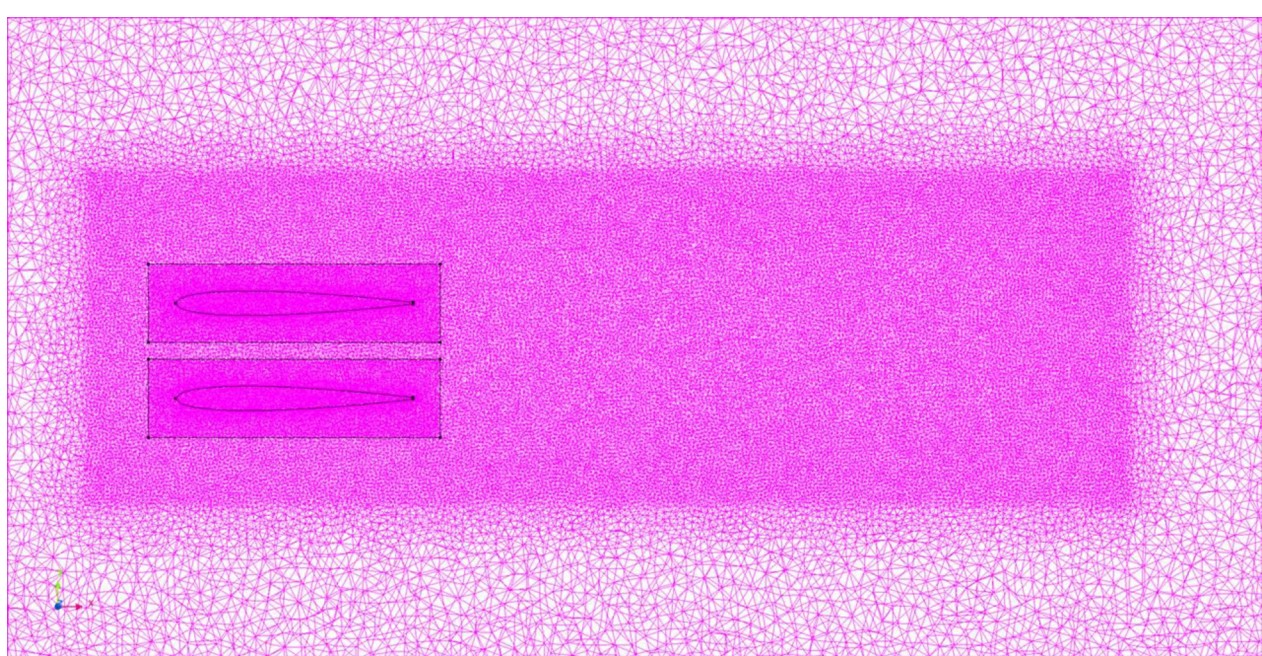

**Fig 6. Mesh partitioning in CFD analysis of parallel swimming.**

Fig 6 shows the mesh partitioning in this analysis. The size of the tetra element was in the range of 2–32 mm, and the size of the model nearest neighbor element was divided such that the tetra element size was 4 mm so that it became finer when approaching the fish body model. The boundary layer elements were inserted to cover the surface of the fish model in eight layers at a rate of change of 1.1 with a thickness of 0.8 mm for the first prism layer.

## Swimming efficiency

We attempted to calculate an index to compare the efficiency during swimming. A measure called the Froude efficiency is often used to assess the swimming efficiency of fish [26,27]. Hemelrijk et al. [17] reported the hydrodynamic advantages of swimming in groups based on the Froude efficiency.

The thrust and lateral forces can be calculated by decomposing the force acting on the model surface into its components. The power of the lateral force can be determined by the product of the lateral component of the velocity of the tail swinging motion and the lateral component of the pressure acting on the surface of the fish body model. The lateral component represents the direction perpendicular to the direction of swimming (the Y direction). In this study, we focused on the surface elements of a fish body model. The ratio of useful power to total power was calculated from the fluid force acting on the surface elements. The swimming efficiency was used as an indicator to compare different conditions.

The internal product of the forces and velocities acting in the direction of travel and in the lateral direction were determined for each element that makes up the surface of the fish body using the pressure distributions calculated from the CFD analysis, as shown in Fig 7. Accordingly, the powers of the thrust and lateral forces were calculated, and the Froude efficiency was obtained for each condition (Eqs 6–8).

$$\eta = \frac{P_T}{P_T + P_S} \tag{6}$$

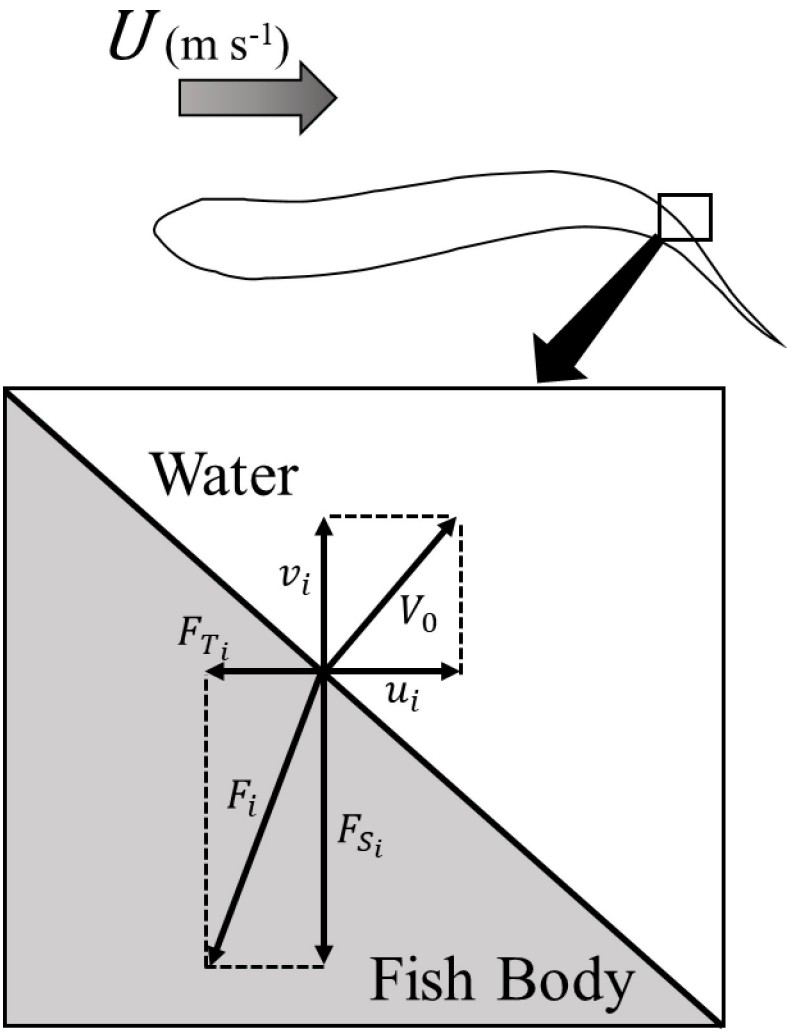

**Fig 7. Decomposition of force $F_i$ on elements of the fish body surface.** $U$ is the inlet flow speed, $V_0$ is the velocity of the element, $u_i$ is the velocity of the element in the direction of travel, $v_i$ is the velocity of the element in the lateral direction, $F_{T_i}$ is the thrust acting on the element, and $F_{S_i}$ is the lateral force acting on the element.

$$P_T = \sum_i F_{T_i} \cdot u_i \tag{7}$$

$$P_S = \sum_i F_{S_i} \cdot v_i \tag{8}$$

where η is the swimming efficiency, $P_T$ is the time-averaged power of the thrust, $P_S$ is the time-averaged power of the lateral force, $F_{T_i}$ is the time-averaged thrust force per element, $F_{S_i}$ is the time-averaged lateral force per element, $u_i$ is the swimming direction velocity of the element, and $v_i$ is the lateral velocity of the element.

We evaluated the swimming efficiency after the thrust and drag values were balanced, as suggested by Borazjani and Sotiropoulos [28].

**Table 2. Constants *a*, *b*, *c*, and *d* in Eq 2 and wavelength (λ) and frequency of caudal fin (*f*) in Eq 3.**

| Swimming Speed (BL s$^{-1}$) | *a* | *b* | *c* | *d* | λ (BL) | *f* (Hz) |
| --- | --- | --- | --- | --- | --- | --- |
| 1.01 | 0.0105 | -5.13 | 0.00440 | 2.69 | 1.00 | 2.00 |
| 1.26 | 0.0175 | -5.30 | 0.00460 | 2.72 | 1.02 | 1.89 |
| 1.52 | 0.0187 | -5.96 | 0.00630 | 2.65 | 1.05 | 2.12 |
| 1.76 | 0.0173 | -4.01 | 0.00490 | 2.73 | 1.03 | 2.55 |
| 2.02 | 0.0235 | -4.51 | 0.00530 | 2.86 | 1.03 | 2.55 |

## Results

### Formulation of swimming motion

The unknown parameters *a*, *b*, *c*, and *d* in Eq 2 were obtained by an exponential approximation using the values of the amplitude at each point on the body axis. Table 2 lists the unknown constants of Eq 2, along with the values of λ and *f* in Eq 3, which were used to determine the swimming motion function at each flow speed. The BL of the swimming speed was based on the body length of the test fish. The time-series data of each point on the body axis of the fish were compared to the measured data and those data obtained from the swimming motion function at a swimming speed of 1.52 BL s$^{-1}$, as shown in Fig 8 ($R^2 = 0.96$).

### CFD analysis of solitary swimming

Fig 9 shows the thrust and drag forces acting on the surface of the fish model during four strokes of tail beating at a swimming speed of 1.52 BL s$^{-1}$. We note a periodicity in the changes in the thrust and drag forces. Moreover, there is a phase difference between the maximum amplitude points at the tip of the caudal fin and the points at which the thrust force are maximum.

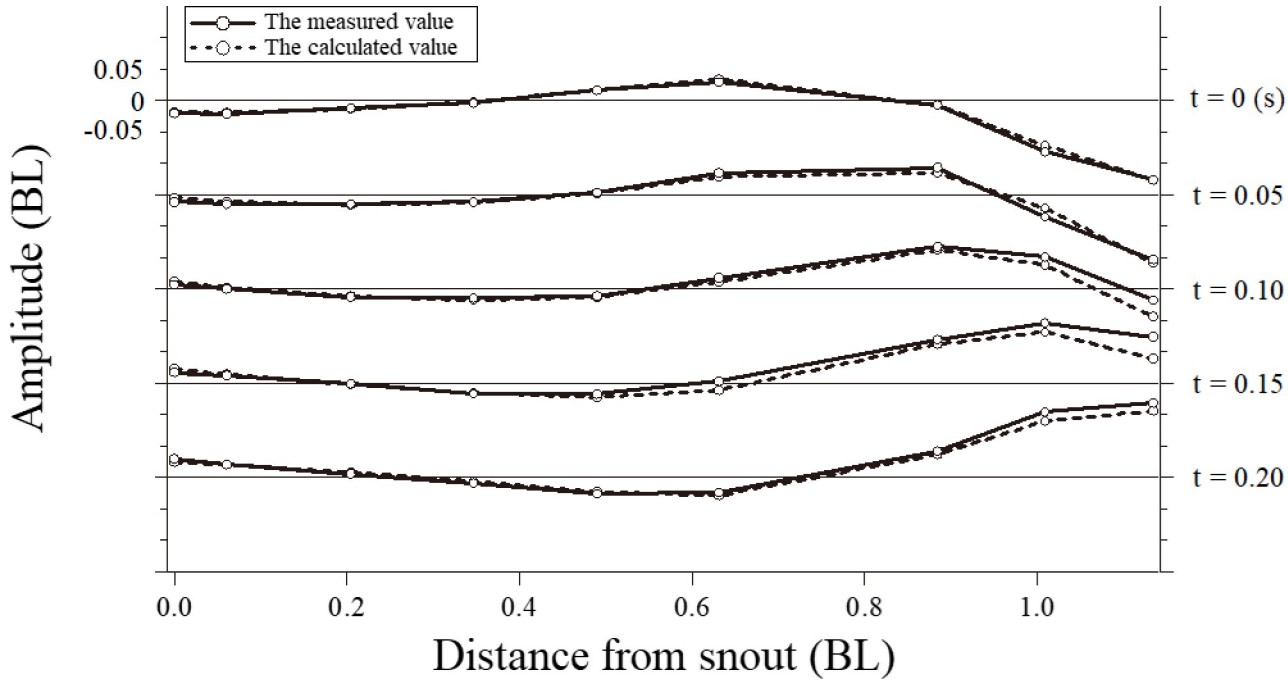

**Fig 8. Time-series of the data for each point on the body axis of the fish.** A comparison is shown between the measured data (solid line) and the results obtained from the swimming motion function (dashed line) at a swimming speed of 1.52 BL s$^{-1}$.

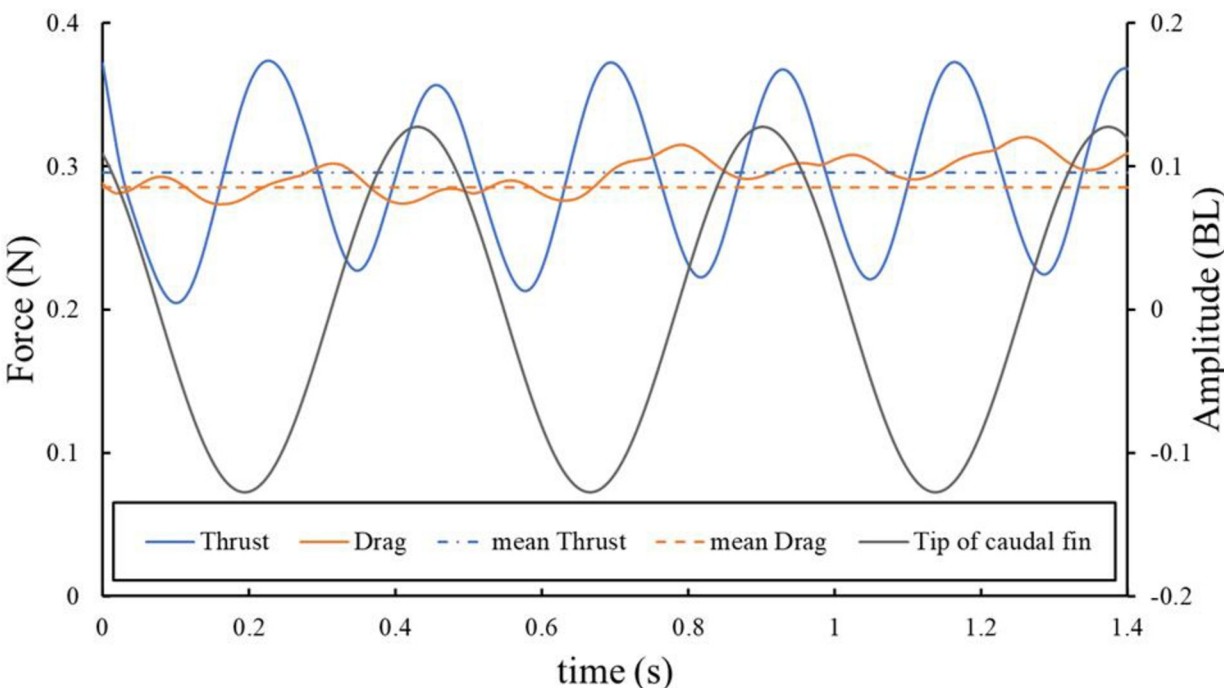

**Fig 9. Thrust and drag acting on the fish model surface.** These forces acted during three strokes of tail beating at a swimming speed of 1.52 BL s⁻¹.

To compare the thrust and drag values, the time-averaged values of the thrust and drag forces were defined as $F_T$ and $F_D$, respectively, and compared at each swimming speed (Table 3). All the swimming motion functions used in the present analysis were determined by conducting a biological experiment in which the test fish steadily swam in a circulatory tank at a constant velocity; thus, the thrust and drag were balanced in this state. At the five swimming speeds analyzed in the CFD calculation, the $F_T/F_D$ ratio was closest to 1 when the swimming speed was 1.52 BL s⁻¹ (Table 3). Fig 10 shows the pressure distributions on the model surface at the maximum speed of the tail tip of the caudal fin at a swimming speed of 1.52 BL s⁻¹.

## CFD analysis of parallel swimming

Fig 11 shows the flow fields around the fish during parallel swimming at individual distances of 0.4 and 0.8L in both inphase and antiphase cases.

In Fig 12, the horizontal axis represents the distance from the origin on a dotted line parallel to the x-axis, with the midpoint of the snout of the two individuals as the origin at an inter-individual distance of 0.8L and under antiphase. As shown in Figs 11 and 12, under both

**Table 3. Time-averaged values of thrust and drag (defined as $F_T$ and $F_D$, respectively) compared at each swimming speed.**

| Swimming Speed (BL s⁻¹) | $F_T$ (N) | $F_D$ (N) | $F_T / F_D$ |
|---|---|---|---|
| 1.01 | 0.201 | 0.133 | 1.51 |
| 1.26 | 0.177 | 0.183 | 0.967 |
| 1.52 | 0.300 | 0.290 | 1.04 |
| 1.76 | 0.351 | 0.373 | 0.942 |
| 2.02 | 0.420 | 0.580 | 0.720 |

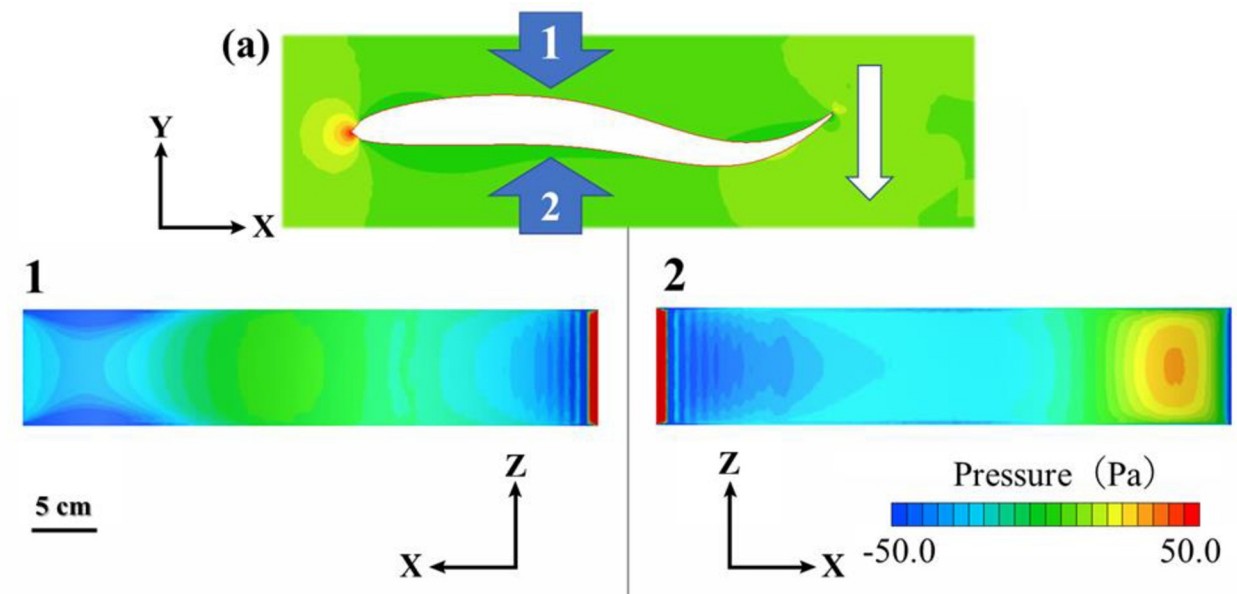

**Fig 10. Model surface pressure distributions.** These show the right (1) and left (2) sides of the caudal fin tail tip at maximum speed (a) at a swimming speed of 1.52 BL s$^{-1}$.

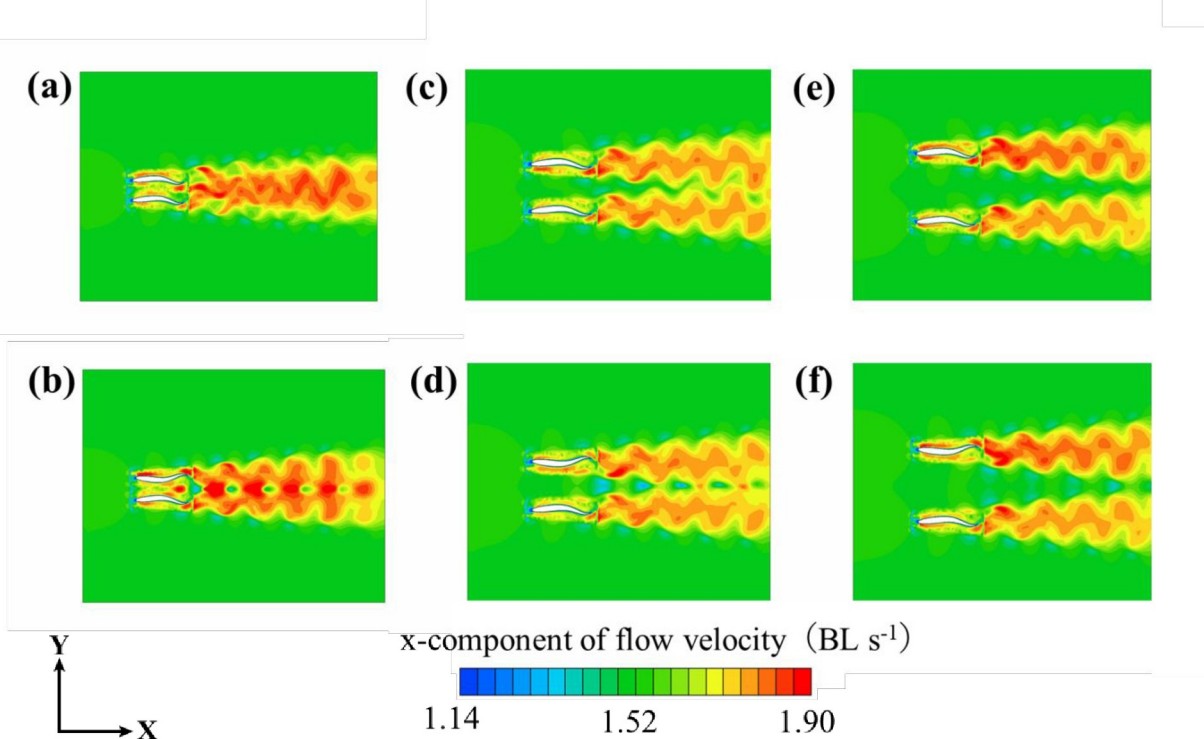

**Fig 11.** Velocity distribution around the model: (a) individual distance of 0.4L and inphase; (b) individual distance of 0.4L and antiphase; (c) individual distances of 0.8L and inphase; (d) individual distance of 0.8L and antiphase; (e) individual distance of 1.2L and inphase; and (f) individual distance of 1.2L and antiphase.

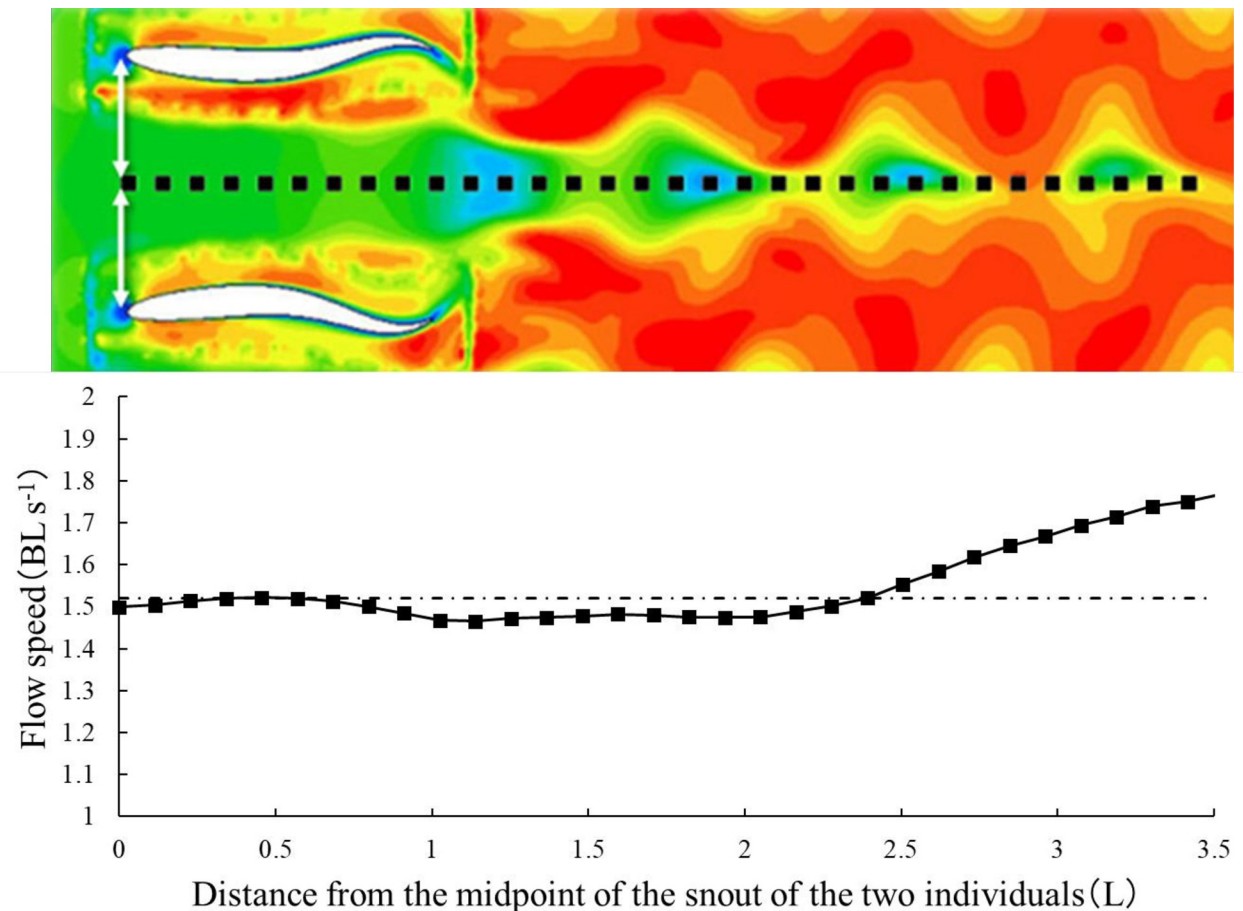

**Fig 12. Velocity value at each point of the velocity distribution.** The horizontal axis represents the distance from the origin on a dotted line parallel to the x-axis with the midpoint of the snout of the two individuals as the origin at an individual distance of 0.8L and under antiphase condition. The vertical axis represents the velocity values on the dotted line. The long-dashed line indicates the inlet flow speed (1.52 BL s$^{-1}$).

inphase and antiphase conditions, at an individual distance of 0.4L, the apparent flow speed in the distance range of 0.5–2.0L from the midpoint of the snout of the two individuals is higher than the inlet flow speed. In contrast, under both phases at inter-individual distances of 0.8L and 1.2L, the apparent flow speed at the aforementioned distance range is lower than the inlet flow speed.

We compared the swimming efficiency of two individuals separated under inphase and antiphase conditions (Fig 13). No significant changes can be found under the inphase conditions. However, under the antiphase conditions (dashed line) and at a distance of 0.4L, the swimming efficiency improved by approximately 10% (Fig 13).

## Discussion

In the pressure distribution of the fish model shown in Fig 10, the pressure at the caudal fin in plane 2 is elevated. Interestingly, on plane 1, an isopleth map similar to that of a caudal peduncle can be observed. Wu [7] reported that the caudal fin is propelled to generate a force, and the aft part of the body must be thin to avoid a negative thrust.

In the wake after the fish bodies, a low-velocity distribution area is generated when the distance between the individuals is 0.8L, as shown in Fig 12. It has been reported that in order to

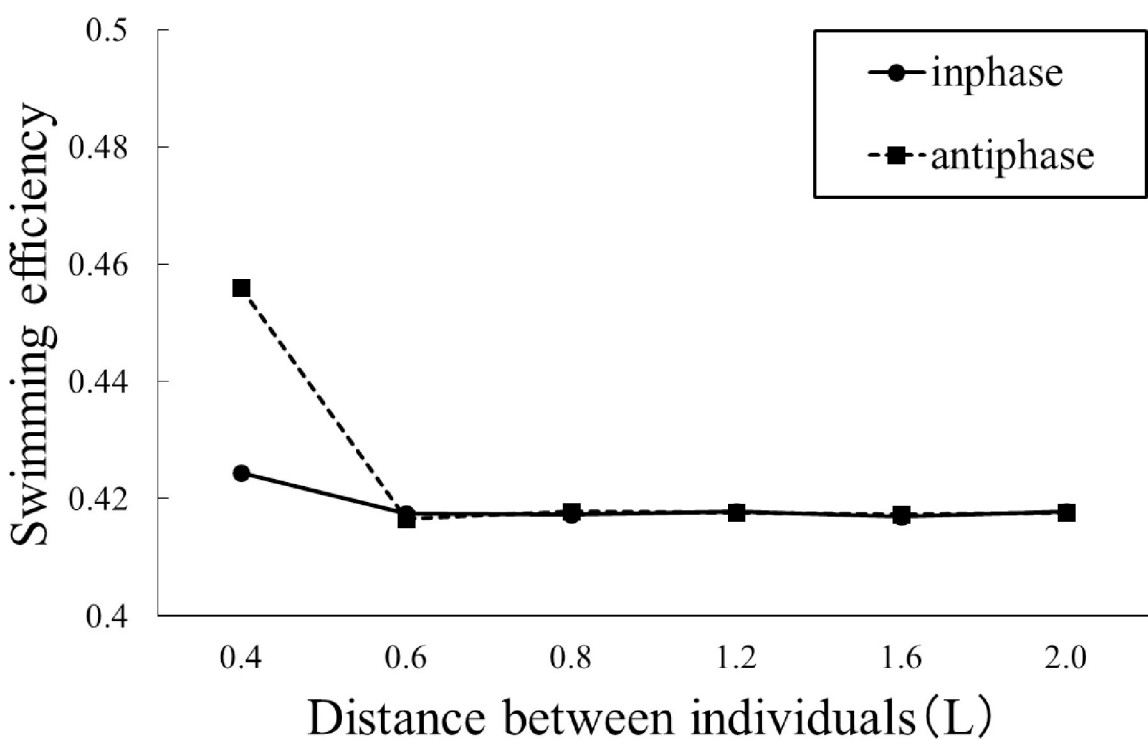

**Fig 13.** Swimming efficiency of two individuals swimming in parallel under inphase (solid line) and antiphase (dashed line) conditions.

achieve maximum savings, the fish should swim continuously in an antiphase tail motion [3,5]. Thus, the apparent flow speed in the wake could be lower than the actual swimming speed when an individual follows the two fish. This implies that the following fish realizes energy savings after a certain distance of travel.

The drag force is proportional to the square of the swimming speed, and the power is proportional to the cube of the swimming speed. When the lateral distance between the individuals was 0.8L and in antiphase, the current velocity of the distance from the snout in the 1–2L range on the midline between two individuals is approximately 0.969 times for one cycle of tail-beating. Therefore, the power required for swimming is approximately 0.910 times, and it is estimated that there is energy-saving effect. The current velocity behind the fish body is 0.974 times, where the distance from the snout is in the range of 1–2L under the inphase condition.

If an individual is at a distance of 1–2L from the snout of the fish, and even if it changed from antiphase to inphase, the fish would have some gain. In contrast, the apparent flow speed was 1.03 times at a distance of 2–3L from the snout even in the antiphase condition. As a school, it is better to have a backward-swimming individual in the distance range of 1–2L, so that the fish can save energy. If the distance between individuals is 1.2L, the actual swimming speed behind the fish is approximately 0.98 times. In this case, the energy savings will not be as high as that when the distance between the individuals is 0.8L. Thus, when the distance between the individuals is 0.8L, it is believed that the following swimmers gain energy. If the distance between the individuals approaches 0.4L, the apparent flow speed could be greater than that of the solitary swimmer, and the load could increase. Therefore, there is no energy saving.

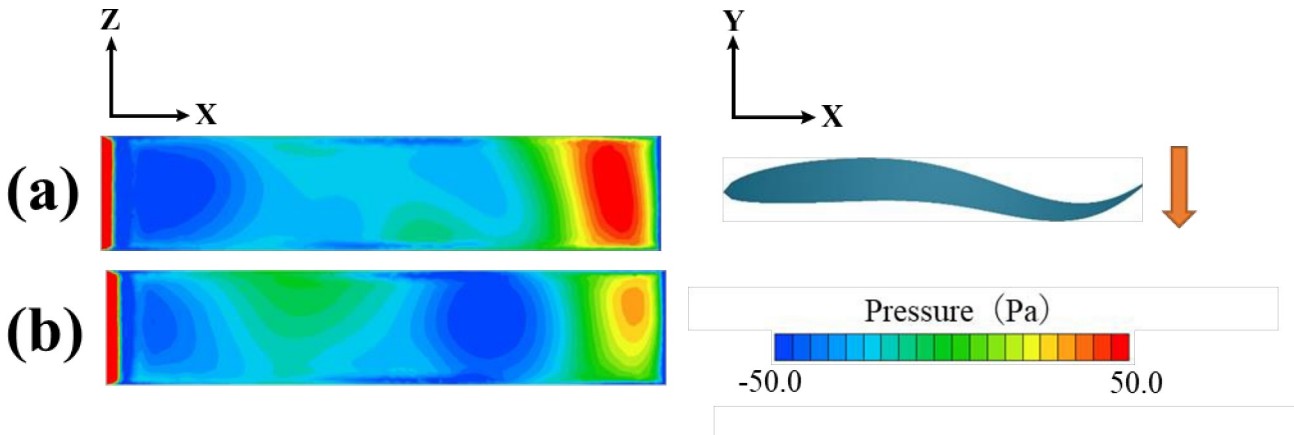

**Fig 14.** Surface pressure contour plot at the maximum velocity of caudal fin amplitude motion: (a) Inter-individual distance of 0.4L, (b) individual distance of 2.0L.

Weihs [5] reported that a diamond-shaped array of fish school is beneficial for increasing the energy savings of the following individuals in theory; however, the value under lateral swimming in this study was approximately 9% of the save rate, and the gain is not as high as that reported in [5]. This may be because Weihs performed the calculation from a simple superposition of vortex arrays by applying the potential theory, which does not adequately reflect the effects of real viscous fluids. This could also be caused by the shape of the 3D body model, as we did not copy the morphology. Verma et al. [29] reported that swimming off-center rather than in a single line can be beneficial.

The swimming efficiency was highest at a distance of 0.4L between the individuals and in the antiphase condition. This may have resulted in a high thrust force owing to the surface pressure contour plot at the maximum velocity of the caudal fin amplitude motion, as shown in Fig 14. Bao et al. [13] reported that two fluttering foils moving in antiphase can induce a Bénard–von Kármán vortex street due to wake interference and improve propulsive performance. A similar phenomenon may have influenced the results of this study. In addition, individuals do not always maintain the phase of the caudal fin oscillation. Therefore, they may repeatedly experience durations of high gain and no gain. However, in terms of the total balance, positive energy gains would be obtained during swimming.

During the live fish experiments, the schools tended to aggregate and synchronize their movements with those of other individuals during fast swimming [30]. This report may support our result that a school of fish with more lateral-side compactness can gain more energy.

Swimming in a fish school is not the only way to save energy. Combining the burst-and-glide swimming [31] and energy saving of each individual can help further increase the total energy saving.

In addition, this study did not consider the acceleration and deceleration of the individuals. It will be necessary to calculate the efficiency considering the acceleration and deceleration of the individuals using CFD analysis in the future.

By further analyzing the 3D structure of schools of fish, we will be able to simulate the fluid dynamics interaction between individuals in fish schools more accurately.

## Acknowledgments

We are grateful to Mr. K. Sasagawa for providing the video images of swimming Biwa Salmon.

## Author Contributions

**Conceptualization:** Tsutomu Takagi.

**Formal analysis:** Keisuke Doi, Tsutomu Takagi.

**Funding acquisition:** Shinsuke Torisawa.

**Project administration:** Shinsuke Torisawa.

**Writing – original draft:** Keisuke Doi.

**Writing – review & editing:** Tsutomu Takagi, Yasushi Mitsunaga, Shinsuke Torisawa.

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
