## [Decision Letter · Decision Letter 0]

18 Nov 2020

PONE-D-20-26743

Hydrodynamical energy-saving analysis of parallelly swimming fish

PLOS ONE

Dear Dr. Torisawa,

Thank you for submitting your manuscript to PLOS ONE. After careful consideration, we feel that it has merit but does not fully meet PLOS ONE’s publication criteria as it currently stands. Therefore, we invite you to submit a revised version of the manuscript that addresses the points raised during the review process.

Please make sure to address the concern of code validation in your response in a detailed manner,.

We look forward to receiving your revised manuscript.

Kind regards,

Roi Gurka

Academic Editor

PLOS ONE

Journal Requirements:

Reviewers' comments:

Reviewer's Responses to Questions

**Comments to the Author**

1. Is the manuscript technically sound, and do the data support the conclusions?

Reviewer #1: Yes

Reviewer #2: Yes

2. Has the statistical analysis been performed appropriately and rigorously? 

Reviewer #1: N/A

Reviewer #2: Yes

3. Have the authors made all data underlying the findings in their manuscript fully available?

Reviewer #1: Yes

Reviewer #2: Yes

4. Is the manuscript presented in an intelligible fashion and written in standard English?

Reviewer #1: Yes

Reviewer #2: No

5. Review Comments to the Author

Reviewer #1: Thanks for the oppertunety to review this paper.

This paper applied the kinematic equations extracted from real fish to CFD to study how fish can improve efficiency in groups. Detail analyses of the conducting and analysing the CFD are given. It is great to see this study improve the accuracy of the CFD by including the real fish's kinematics. The main issues of this current versions are:

1. In the abstract and title claim that this paper talks about the energy-saving, however, most of the paper focused on the CFD method and verifications of the CFD results. The report of the energy-saving in schools is very limited and simple. The mechanism of energy saving (gaining higher efficiency reported in this paper) is not given. There are many papers report the hydrodynamic interaction mechanism of energy saving in schooling fish swimming side-by-side. This paper did not have a detail comparisons, discussions and analyses with them. e.g.

Huertas-Cerdeira, C., B. Fan, and M. Gharib, Coupled motion of two side-by-side inverted flags. Journal of Fluids and Structures, 2018. 76: p. 527-535.

Bao, Y., et al., Dynamic interference of two anti-phase flapping foils in side-by-side arrangement in an incompressible flow. Physics of Fluids, 2017. 29(3): p. 033601.

Dewey, P.A., et al., Propulsive performance of unsteady tandem hydrofoils in a side-by-side configuration. Physics of Fluids, 2014. 26(4): p. 041903.

Li, G., et al., On the energetics and stability of a minimal fish school. PLoS ONE, 2019. 66(2): p. 596023.

Dong, G.-J. and X.-Y. Lu, Characteristics of flow over traveling wavy foils in a side-by-side arrangement. Physics of Fluids, 2007. 19(5): p. 057107.

2. Another main issue is the writing. I recommend the authors rewrite the title and abstract to summarise this paper's work accurately, and reorganise the paragraphs according the to the nice logic. For example, Page 18 Line 291-294, those body wave comparisons should be given near the Section Formulation of swimming motion. Introduction should be more related to the novelty in this paper, (to me it is applying real fish kinematics to CFD, if the authors give the new mechanism of energy-saving, then it should be this).

minor comments:

1. An extral space at "Biwa salmon ( Oncorhynchus)" in the abstract in the system.

2. Page 2 Line 21-25, sentence is too long.

3. Page 5 Line 66 It says "the purpose of this study was to clarify the effect of parallel swimming in a group ...", this seems different from the title and abstract. Please change to keep consistent.

4. Table. 1, how many fish are measured? And move the description of the permit at Page 16 Line 262 here, as these should be given in the methods instead of CFD results.

5. Page 7 Eqs. 1-3, clarify the unit of the varibles.

6. Page 8 Line 121 what is the unit of 0.25? BL or meter?

7. Page 8 Line 123, Should "x<0.25" be "x>0.25"?

8. Page 14 Line 214, report Renoylds number

9. Page 14 Line 222, report flow speed in CFD

10. Page 16 Line 256, you also need to point out Froude efficiency mainly works if the fish is accelerating or decelerating. See a nice paper discussed this problem: Maertens, A.P., M.S. Triantafyllou, and D.K.P. Yue, Efficiency of fish propulsion. Bioinspiration & Biomimetics, 2015. 10(4): p. 046013.

11. Page 17 Line 274 Please also report the R value for the fitting. or any other values which can tell how good the fitting is.

12. Page 18 Line 280, from which figure we can get this claim? What is the phase shift value?

13. Page 18 Line 291 add the table ref or the value for the claim "was closest to 1"

12. Page 20 Line 320, new paragraph after the caption of the figure

13. Page 21 Line 328, report which line shows the "10%" energy saving?

14. Page 21 Line 339-341, this is the point of Weihs (1973 Nature) paper, please cite. Also, you need to mention this is valid when two fish swim out-of-phase.

15. Page 23 Line 366, this can also be due to the 3D body shape as you did not copy the morphology. And 3D effect of simulation can also be an explanation. See this paper, Verma, S., G. Novati, and P. Koumoutsakos, Efficient collective swimming by harnessing vortices through deep reinforcement learning. Proceedings of the National academy of Sciences of the United States of America, 2018. 115(23): p. 5849-5854.

16. Page 24 Line 383 what are the "new insights"?

Check the references carefully. Some have dois (e.g. [1]), some do not (e.g. [2]). Some even have different font size (e.g. [11]).

Fig.1 give information of the view angle, side view or top view?

Fig.9 give line legend

Fig. 11 add the coordinate to show the direction of x

Fig. 13 line legend is missing

Reviewer #2: The study does not provide the non-dimensional number such as Reynolds number and Strouhal number

There are also grammatical and spelling mistakes in the manuscript.

There are no validations for the code

6. PLOS authors have the option to publish the peer review history of their article (what does this mean?). If published, this will include your full peer review and any attached files.

Reviewer #1: No

Reviewer #2: No

---

## [Author Response · Author response to Decision Letter 0]

29 Jan 2021

Dear Editor and Reviewers:

Thank you very much for your helpful suggestions and comments. We are enclosing a revised version of our manuscript, along with our responses to your comments and suggestions regarding the document.

The manuscript has been carefully rechecked and appropriate changes have been made in accordance with the suggestions received. The major changes made are highlighted in yellow in the revised document. The responses to your comments have also been prepared and are attached herewith.

We thank you for your thoughtful suggestions and insights, which have enriched the manuscript and produced a more balanced and better account of the research. We hope that the revised manuscript is now suitable for publication in PLOS ONE.

Thank you for your consideration. We look forward to hearing from you.

Yours sincerely,

Shinsuke Torisawa

 

To Reviewer #1

1. In the abstract and title claim that this paper talks about the energy-saving, however, most of the paper focused on the CFD method and verifications of the CFD results. The report of the energy- saving in schools is very limited and simple. The mechanism of energy saving (gaining higher efficiency reported in this paper) is not given. There are many papers report the hydrodynamic interaction mechanism of energy saving in schooling fish swimming side-by-side. This paper did not have a detail comparisons, discussions and analyses with them. e.g.

Huertas-Cerdeira, C., B. Fan, and M. Gharib, Coupled motion of two side-by-side inverted flags. Journal of Fluids and Structures, 2018. 76: p. 527-535.

Bao, Y., et al., Dynamic interference of two anti-phase flapping foils in side-by-side arrangement in an incompressible flow. Physics of Fluids, 2017. 29(3): p. 033601.

Dewey, P.A., et al., Propulsive performance of unsteady tandem hydrofoils in a side-by-side configuration. Physics of Fluids, 2014. 26(4): p. 041903.

Li, G., et al., On the energetics and stability of a minimal fish school. PLoS ONE, 2019. 66(2): p. 596023.

Dong, G.-J. and X.-Y. Lu, Characteristics of flow over traveling wavy foils in a side-by-side arrangement. Physics of Fluids, 2007. 19(5): p. 057107.

Response

Thank you for your useful comments. Accordingly, we have changed the title from “Hydrodynamical energy-saving analysis of parallelly swimming fish” to “Hydrodynamical effect of parallelly swimming fish by using CFD method.” We have also reviewed added the references you suggested (Page 4 Lines 48-49), and added to the Discussion section (Page 24 Lines 380-383).

2. Another main issue is the writing. I recommend the authors rewrite the title and abstract to summarise this paper's work accurately, and reorganise the paragraphs according the to the nice logic. For example, Page 18 Line 291-294, those body wave comparisons should be given near the Section Formulation of swimming motion. Introduction should be more related to the novelty in this paper, (to me it is applying real fish kinematics to CFD, if the authors give the new mechanism of energy-saving, then it should be this).

Response

Thank you for your useful comments. In response to your suggestion, we have changed the title as we mentioned above. 

We moved the “The time-series data of each point on the body axis of the fish were compared between the measured data and that obtained from the swimming motion function at a swimming speed of 1.52 BL s−1, as shown in Fig 9.” mentioned in the CFD analysis to the Section “Formulation of swimming motion” (Page 17 Lines 276-279).

We have added “In other words, we applied the real fish kinematics to CFD analysis.” to Introduction (Page 5 Lines 75-76). The abstract was revised to state “with real fish swimming motion” (Page 2 Line 16).

minor comments:

1. An extral space at "Biwa salmon ( Oncorhynchus)" in the abstract in the system.

Response

Thank you. It has been corrected.

2. Page 2 Line 21-25, sentence is too long.

Response

Thank you for your useful comment. We have revised “Page 2 Lines 21-25” to split it into two sentences.

3. Page 5 Line 66 It says "the purpose of this study was to clarify the effect of parallel swimming in a group ...", this seems different from the title and abstract. Please change to keep consistent.

Response

Thank you for the suggestion; we have changed the title from “Hydrodynamical energy-saving analysis of parallelly swimming fish” to “Hydrodynamical effect of parallelly swimming fish by using CFD method.”

4. Table. 1, how many fish are measured? And move the description of the permit at Page 16 Line 262 here, as these should be given in the methods instead of CFD results.

Response

Thank you for this comment. We experimented with one test fish over five different flow rates. We have also moved the permit text to “Page 9 Lines 128-130.”

5. Page 7 Eqs. 1-3, clarify the unit of the varibles.

Response

We appreciate your comment on this point. We have indicated the units as you pointed out (Page 8 Lines 113-115).

6. Page 8 Line 121 what is the unit of 0.25? BL or meter?

Response

We appreciate your comment on this point. We have indicated the units as you pointed out (Page 8 Line 113).

7. Page 8 Line 123, Should "x<0.25" be "x>0.25"?

Response

Thank you for this comment. The text has been revised to explain correctly the method of fitting function forms (Pages 8-9 Lines 122-127).

8. Page 14 Line 214, report Renoylds number

Response

Thank you for your useful comment. We have denoted the Reynolds number (Page 14 Lines 218-219).

9. Page 14 Line 222, report flow speed in CFD

Response

Thank you for your useful comment. We have denoted the flow speed (Page 14 Lines 227-228).

10. Page 16 Line 256, you also need to point out Froude efficiency mainly works if the fish is accelerating or decelerating. See a nice paper discussed this problem: Maertens, A.P., M.S. Triantafyllou, and D.K.P. Yue, Efficiency of fish propulsion. Bioinspiration & Biomimetics, 2015. 10(4): p. 046013.

Response

We appreciate your comment on this point. The following text has been added to the discussion section. “In addition, this study did not consider the acceleration and deceleration of the individuals. It will be necessary to calculate the efficiency considering the acceleration and deceleration of the individuals using CFD analysis in the future.” (Page 25 Lines 397-399)

11. Page 17 Line 274 Please also report the R value for the fitting. or any other values which can tell how good the fitting is.

Response

Thank you for your useful comment. We have added coefficient of determination (Page 17 Line 279).

12. Page 18 Line 280, from which figure we can get this claim? What is the phase shift value?

Response

Thank you for this question. We have modified and indicated the figure (Fig 9).

13. Page 18 Line 291 add the table ref or the value for the claim "was closest to 1"

Response

Thank you for your useful comment. We have attached a table reference (Page 19 Line 302).

14. Page 20 Line 320, new paragraph after the caption of the figure

Response

We are uncertain as to the exact section being referred to. However, we have changed the font to make the figure caption clearer.

15. Page 21 Line 328, report which line shows the "10%" energy saving?

Response

Thank you for this question. We have added a figure legend and specified that the value of the inverse phase is the dashed line (Page 21 Lines 335-336).

16. Page 21 Line 339-341, this is the point of Weihs (1973 Nature) paper, please cite. Also, you need to mention this is valid when two fish swim out-of-phase.

Response

Thank you for this comment. We have reviewed and cited the paper (Weihs, 1973;1975). We have also added it to the discussion section (Page 22 Lines 345-347).

17. Page 23 Line 366, this can also be due to the 3D body shape as you did not copy the morphology. And 3D effect of simulation can also be an explanation. See this paper, Verma, S., G. Novati, and P. Koumoutsakos, Efficient collective swimming by harnessing vortices through deep reinforcement learning. Proceedings of the National academy of Sciences of the United States of America, 2018. 115(23): p. 5849-5854.

Response

We highly appreciate your suggestion on this point. We have added it and cited it in the discussion section (Pages 23-24 Lines 374-376).

18. Page 24 Line 383 what are the "new insights"?

Response

Thank you for this question. We have revised and added that information to the discussion section (Page 25 Lines 400-401).

Check the references carefully. Some have dois (e.g. [1]), some do not 

(e.g. [2]). Some even have different font size (e.g. [11]).

Response

Thank you for your useful comment. For those that have been assigned a doi, it is listed, but for those that are unknown (e.g. [2]), it is not. We have also fixed the font size.

Fig.1 give information of the view angle, side view or top view?

Response

Thank you for your useful comment. It is top view. We have revised Fig 1.

Fig.9 give line legend

Response

Thank you for your useful comment. We have revised Fig 8 (Previous Fig 9).

Fig. 11 add the coordinate to show the direction of x

Response

Thank you for your useful comment. We have revised Fig 11.

Fig. 13 line legend is missing

Response

Thank you for your useful comment. We have revised Fig 13

 

Reviewer #2: 

The study does not provide the non-dimensional number such as Reynolds number and Strouhal number

Response

We highly appreciate your comment on this point. The Reynolds number has been added to the text (Page 14 Line 219). It is 2.60×105. The Strouhal number of the 3D fish model was 0.19 to 0.20.

There are also grammatical and spelling mistakes in the manuscript.

Response

Thank you for this comment. The revised paper has been proofread by a professional English language editing service, with experienced scientific editors. This has improved the grammar and stylistic expression of the paper.

There are no validations for the code

Response

Thank you for this comment. We have added the reference (Page 10 Line 149). Please refer to the following study, which confirms the calculation results of the software used in this study:

Shiino Y, Kuwazuru O. Comparative experimental and simulation study on passive feeding flow generation in 'Cyrtospirifer'. Memoirs of the Association of Australasian Palaeontologists. 2011:1-8.

Best regards,

Shinsuke Torisawa

---

## [Decision Letter · Decision Letter 1]

12 Feb 2021

PONE-D-20-26743R1

Hydrodynamical effect of parallelly swimming fish by using CFD method

PLOS ONE

Dear Dr. Torisawa,

Thank you for submitting your manuscript to PLOS ONE. After careful consideration, we feel that it has merit but does not fully meet PLOS ONE’s publication criteria as it currently stands. Therefore, we invite you to submit a revised version of the manuscript that addresses the points raised during the review process.

Please address adequately the reviewer concern and make a proper connection between the simulations, fish and previous research work in this field.

We look forward to receiving your revised manuscript.

Kind regards,

Roi Gurka

Academic Editor

PLOS ONE

Reviewers' comments:

Reviewer's Responses to Questions

**Comments to the Author**

1. If the authors have adequately addressed your comments raised in a previous round of review and you feel that this manuscript is now acceptable for publication, you may indicate that here to bypass the “Comments to the Author” section, enter your conflict of interest statement in the “Confidential to Editor” section, and submit your "Accept" recommendation.

Reviewer #1: (No Response)

Reviewer #2: All comments have been addressed

2. Is the manuscript technically sound, and do the data support the conclusions?

Reviewer #1: Yes

Reviewer #2: Yes

3. Has the statistical analysis been performed appropriately and rigorously? 

Reviewer #1: Yes

Reviewer #2: Yes

4. Have the authors made all data underlying the findings in their manuscript fully available?

Reviewer #1: Yes

Reviewer #2: Yes

5. Is the manuscript presented in an intelligible fashion and written in standard English?

Reviewer #1: (No Response)

Reviewer #2: Yes

6. Review Comments to the Author

Reviewer #1: Dear editor and authors,

Thanks for the opportunity to review this paper. And thank the authors addressed my most comments. But sorry, I can not accept as this because I still do not think the reviewer did nicely summarise the major contribution of the work.

I do not agree that the major contribution is “applied the real fish kinematics to CFD analysis ”. Because the authors did not really apply the kinematics of real fish frame by frame. Instead, they extract the key parameters from the real fish system and then simulated by the fitted body wave function. This is done by many recent works such as:

Li, G., Kolomenskiy, D., Liu, H., Thiria, B. and Godoy-Diana, R., 2019. On the energetics and stability of a minimal fish school. PloS one, 14(8), p.e0215265.

Li, L., Nagy, M., Graving, J.M., Bak-Coleman, J., Xie, G. and Couzin, I.D., 2020. Vortex phase matching as a strategy for schooling in robots and in fish. Nature communications, 11(1), pp.1-9.

And the authors mentioned that “there are no published studies on the calculation of the thrust and drag forces from the pressure distribution on the body …” However, the truth is there are many. For example:

Dabiri, J.O., Bose, S., Gemmell, B.J., Colin, S.P. and Costello, J.H., 2014. An algorithm to estimate unsteady and quasi-steady pressure fields from velocity field measurements. Journal of Experimental Biology, 217(3), pp.331-336.

Gemmell, B.J., Colin, S.P., Costello, J.H. and Dabiri, J.O., 2015. Suction-based propulsion as a basis for efficient animal swimming. Nature communications, 6(1), pp.1-8.

Lentink, D., Haselsteiner, A.F. and Ingersoll, R., 2015. In vivo recording of aerodynamic force with an aerodynamic force platform: from drones to birds. Journal of The Royal Society Interface, 12(104), p.20141283.

I would really suggest the authors do read the related literatures and summary nicely about the contribution in the abstract and introduction.

Reviewer #2: Dear Writer

The strouhal number should be in the manuscript. It is not mentioned in the manuscript.

7. PLOS authors have the option to publish the peer review history of their article (what does this mean?). If published, this will include your full peer review and any attached files.

Reviewer #1: **Yes: **Liang Li

Reviewer #2: No

---

## [Author Response · Author response to Decision Letter 1]

13 Apr 2021

Dear Editor and Reviewers:

Thank you very much for your helpful comments. We are enclosing a revised version of our manuscript, along with our responses to your suggestions regarding the document.

The manuscript has been carefully rechecked and appropriate changes have been made in accordance with the suggestions received. The major changes made are highlighted in yellow in the revised document. Detailed responses to all comments have also been prepared and are attached herewith.

We thank you for your thoughtful suggestions and insights, which have enriched the manuscript and produced a more balanced and better account of the research. We hope that the revised manuscript is now suitable for publication in PLOS ONE.

Thank you for your consideration. We look forward to hearing from you.

Yours sincerely,

Shinsuke Torisawa

To Reviewer #1

Q1. I do not agree that the major contribution is "applied the real fish kinematics to CFD analysis". Because the authors did not really apply the kinematics of real fish frame by frame. Instead, they extract the key parameters from the real fish system and then simulated by the fitted body wave function. This is done by many recent works such as:

Li, G., Kolomenskiy, D., Liu, H., Thiria, B. and Godoy-Diana, R., 2019. On the energetics and stability of a minimal fish school. PloS one, 14(8), p.e0215265.

Li, L., Nagy, M., Graving, J.M., Bak-Coleman, J., Xie, G. and Couzin, I.D., 2020. Vortex phase matching as a strategy for schooling in robots and in fish. Nature communications, 11(1), pp.1-9.

Response

A1. Thank you for your useful comments. The following text has been added to the abstract: “There are limited examples of fish models based on actual swimming movements using simulation, and the movements in these models are simple.” (Page 2 Lines 12-13) 

We have also cited the previous study of Li et al. (2019 [16]) that Reviewer #1 suggested, and the following text has been added to the introduction section. “There are several examples of body wave functions applied to CFD, but their lateral excursion is simple. The body wave functions used had amplitudes such that the simple envelopes became larger as they approached the tail tip [16, 17].” (Page 6 Lines 81-82)

In addition, the following text has been added to the introduction section. “Using a pair of swimming fish robots, Li et al. [9] hypothesized that energy savings could be achieved using vortex phase-matching.” (Pages 3-4 Lines 42-44)

Q2. And the authors mentioned that “there are no published studies on the calculation of the thrust and drag forces from the pressure distribution on the body”; However, the truth is there are many. For example:

Dabiri, J.O., Bose, S., Gemmell, B.J., Colin, S.P. and Costello, J.H., 2014. An algorithm to estimate unsteady and quasi-steady pressure fields from velocity field measurements. Journal of Experimental Biology, 217(3), pp.331-336.

Gemmell, B.J., Colin, S.P., Costello, J.H. and Dabiri, J.O., 2015. Suction-based propulsion as a basis for efficient animal swimming. Nature communications, 6(1), pp.1-8.

Lentink, D., Haselsteiner, A.F. and Ingersoll, R., 2015. In vivo recording of aerodynamic force with an aerodynamic force platform: from drones to birds. Journal of The Royal Society Interface, 12(104), p.20141283.

I would really suggest the authors do read the related literatures and summary nicely about the contribution in the abstract and introduction.

Response

A2. Thank you for your useful comments and introducing these useful papers. The following text has been added to the introduction section. “Dabiri et al. [19] used an algorithm to determine the pressure distribution from the velocity field to obtain the pressure coefficient. Hemelrijk et al. [17] used multi-particle collision dynamics (MPCD) to estimate the thrust and drag forces acting on a fish body model. Limited work has been done to determine the thrust and drag forces acting on a fish body model using these methods.” (Pages 4-5 Lines 60-64)

However, Lentink et al. (2015) proposed a methodology of measuring thrust in a non-invasive manner, and Gemmell et al. (2015) focused on the functionality of thrust generation in aquatic animals such as jellyfish that swim mainly by suction. Therefore, we think it appropriate not to cite these two papers.

To Reviewer #2: 

Q1. The strouhal number should be in the manuscript. It is not mentioned in the manuscript.

Response

A1. We highly appreciate your comment on this point. The Strouhal number of the 3D fish model has been estimated and is given in the revised manuscript. (Pages 12 Line 182-184, Page 14 Lines 231-232)

Changes other than those the reviewers pointed out:

 The abbreviation (CFD) in the title of the paper has been changed to computational fluid dynamics. Thus, the title has been updated: "Hydrodynamical effect of parallelly swimming fish using computational fluid dynamics method"

The ethics statement was replaced Page 9 Line129-131 into Page 6 Lines 89-91 as the staff of PLOS ONE Office suggested.

---

## [Decision Letter · Decision Letter 2]

15 Apr 2021

Hydrodynamical effect of parallelly swimming fish using computational fluid dynamics method

PONE-D-20-26743R2

Dear Dr. Torisawa,

We’re pleased to inform you that your manuscript has been judged scientifically suitable for publication and will be formally accepted for publication once it meets all outstanding technical requirements.

Kind regards,

Roi Gurka

Academic Editor

PLOS ONE

Additional Editor Comments (optional):

Reviewers' comments:

Reviewer's Responses to Questions

**Comments to the Author**

1. If the authors have adequately addressed your comments raised in a previous round of review and you feel that this manuscript is now acceptable for publication, you may indicate that here to bypass the “Comments to the Author” section, enter your conflict of interest statement in the “Confidential to Editor” section, and submit your "Accept" recommendation.

Reviewer #1: All comments have been addressed

2. Is the manuscript technically sound, and do the data support the conclusions?

Reviewer #1: Yes

3. Has the statistical analysis been performed appropriately and rigorously? 

Reviewer #1: N/A

4. Have the authors made all data underlying the findings in their manuscript fully available?

Reviewer #1: Yes

5. Is the manuscript presented in an intelligible fashion and written in standard English?

Reviewer #1: Yes

6. Review Comments to the Author

Reviewer #1: I thank the authors for their answers to my comments. The quality has been greatly improved. Therefore, I support to publish.

7. PLOS authors have the option to publish the peer review history of their article (what does this mean?). If published, this will include your full peer review and any attached files.

Reviewer #1: **Yes: **Liang Li

---

## [Editor Report · Acceptance letter]

19 Apr 2021

PONE-D-20-26743R2 

Hydrodynamical effect of parallelly swimming fish using computational fluid dynamics method 

Dear Dr. Torisawa:

I'm pleased to inform you that your manuscript has been deemed suitable for publication in PLOS ONE. Congratulations! Your manuscript is now with our production department. 

Kind regards, 

on behalf of

Dr. Roi Gurka 

Academic Editor

PLOS ONE